# Epidemiology of *Burkholderia pseudomallei, Streptococcus suis, Salmonella* spp., *Shigella* spp. and *Vibrio* spp. infections in 111 hospitals in Thailand, 2022

Charuttaporn Jitpeera[1], Somkid Kripattanapong[1], Preeyarach Klaytong[2], Chalida Rangsiwutisak[2], Prapass Wannapinij[2], Pawinee Doungngern[1], Papassorn Pinyopornpanish[3], Panida Chamawan[4], Voranadda Srisuphan[4], Krittiya Tuamsuwan[4], Phairam Boonyarit[4], Orapan Sripichai[5], Soawapak Hinjoy[6], John Stelling[7], Paul Turner[2,8,9], Wichan Bhunyakitikorn[1], Sopon Iamsirithaworn[10], Direk Limmathurotsakul[2,9,11]*

1 Division of Epidemiology, Department of Disease Control, Ministry of Public Health, Nonthaburi, Thailand, 2 Mahidol-Oxford Tropical Medicine Research Unit, Faculty of Tropical Medicine, Mahidol University, Bangkok, Thailand, 3 Division of Communicable Diseases, Department of Disease Control, Ministry of Public Health, Nonthaburi, Thailand, 4 Health Administration Division, The Office of Permanent Secretary, Ministry of Public Health, Nonthaburi, , 5 Department of Medical Science, Ministry of Public Health, Nonthaburi, Thailand, 6 Department of Disease Control, Ministry of Public Health, Nonthaburi, Thailand, 7 Brigham and Women's Hospital and Harvard Medical School, Boston, Massachusetts, United States of America, 8 Cambodia-Oxford Medical Research Unit, Angkor Hospital for Children, Siem Reap, Cambodia, 9 Centre for Tropical Medicine and Global Health, University of Oxford, Oxford, United Kingdom, 10 Inspection Division, Department of Disease Control, Ministry of Public Health, Nonthaburi, Thailand, 11 Department of Tropical Hygiene, Faculty of Tropical Medicine, Mahidol University, Bangkok, Thailand

* direk@tropmedres.ac

**Data availability statement:** The hospital-level summary data used for the study are open-access and available at https://doi.org/10.6084/m9.figshare.25907494

## Abstract

The information on notifiable diseases in low- and middle-income countries is often incomplete, limiting our understanding of their epidemiology. Our study addresses this knowledge gap by analyzing microbiology laboratory and hospital admission data from 111 of 127 public referral hospitals in Thailand, excluding Bangkok, from January to December 2022. We evaluated factors associated with the incidence of notifiable bacterial diseases (NBDs) caused by 11 pathogens; including *Brucella* spp., *Burkholderia pseudomallei*, *Corynebacterium diphtheriae*, *Neisseria gonorrhoeae*, *Neisseria meningitidis*, non-typhoidal *Salmonella* spp. (NTS), *Salmonella enterica* serovar Paratyphi, *Salmonella enterica* serovar Typhi, *Shigella* spp., *Streptococcus suis*, and *Vibrio* spp.. We used multivariable Poisson random-effects regression models. Additionally, we compared their yearly incidence rates in 2022 with those from 2012-2015 in hospitals where paired data were available. In 2022, the NBD associated with the highest total number of deaths was *B. pseudomallei* (4,407 patients; 1,219 deaths) infection, followed by NTS (4,501 patients; 461 deaths), *S. suis* (867 patients, 134 deaths) and *Vibrio* spp. (809 patients, 122 deaths) infection. The incidence rates of *B. pseudomallei, S. suis* and *Vibrio* spp. infections were highest in the northeast, upper central and west, respectively. The incidence rate of NTS infection was generally high across all geographical regions. The yearly incidence rates of *B. pseudomallei* and *S. suis* infections in 2022 were higher than those between

**Funding:** This research was funded in whole, or in part, by the Wellcome Trust (Grant 224681/Z/21/Z to DL). For the purpose of Open Access, the author has applied a CC BY public copyright licence to any Author Accepted Manuscript version arising from this submission. The funders had no role in study design, data collection and analysis, decision to publish, or preparation of the manuscript.

**Competing interests:** The authors have declared that no competing interests exist.

2012-2015, while those of fecal-oral transmitted NBDs including NTS infection, typhoid, shigellosis and vibriosis were lower. Overall, *B. pseudomallei* and *S. suis* infections are emerging and associated with a very high number of deaths in Thailand. Although the incidence of NTS infection and vibriosis are decreasing, they are still associated with a high number of cases and deaths. Specific public health interventions are warranted.

## Introduction

Timely, reliable and complete information regarding notifiable diseases is essential for disease control and prevention, and enhancing our understanding of their epidemiology [1–3]. To achieve timeliness and completeness in data reporting, many high-income and upper-middle-income countries have strengthened their national surveillance systems by modernizing tools, technology and strategies [4–6]. These include automatic electronic laboratory-based reporting of notifiable diseases [5–7]. However, most low and middle-income countries (LMICs) have a shortage of resources, still use conventional or semi-automated data reporting systems, and do not automatically integrate laboratory data into their surveillance systems [4,8]. Therefore, the data available in the national surveillance systems of LMICs are still largely incomplete [9–11] and our understanding of their epidemiology remains limited.

In Thailand, the national surveillance systems monitors 13 dangerous communicable diseases and 57 notifiable diseases, overseen by the Department of Disease Control (DDC), Ministry of Public Health (MoPH) [12]. Of 13 dangerous communicable diseases, 11 were viral infections (e.g., Ebola infection) and two were bacterial infections (*Yersinia pestis* infection and extensively drug-resistant *Mycobacterium tuberculosis*). Immediate reporting of any suspected cases of these 13 diseases is required. For the 57 notifiable diseases, the reporting systems can be semi-automatic, utilizing the electronic data of final diagnosis based on the International Classification of Diseases, 10th revision (ICD-10) recorded in the hospital information systems (HIS). However, the ICD-10 is reliable only in few conditions [13,14] and the surveillance systems do not utilize microbiology laboratory data. The ICD-10 is unreliable for notifiable bacterial diseases (NBDs) because many disease diagnoses recorded by physicians are incomplete, ICD-10 coders enter codes based solely on the diagnoses recorded by physicians, and coders tend to code only diagnoses that result in reimbursement [13,14]. Recent studies showed that many public hospitals, which have microbiology laboratories, do not report most cases and deaths following culture-confirmed NBDs to the national surveillance systems [11,15]. The incomplete data hinders efforts to improve awareness, control, prevention, and our understanding of NBDs in the country [16].

To overcome limitations in LMICs, we developed the AMASS (AutoMated tool for Antimicrobial resistance Surveillance System), an offline application that enables hospitals to automatically analyse and generate standardized antimicrobial resistance (AMR) surveillance reports from routine microbiology and hospital data [17]. The AMASS version 1.0 was released on 1st February 2019 and tested in seven hospitals in seven countries [17]. We conceptualized that the AMASS could additionally analyse and generate summary reports for multiple NBDs [18]. The AMASS version 2.0 (AMASSv2.0) was released on 16th May 2022 and tested using data of 49 public hospitals in Thailand from 2012 to 2015 [15]. We demonstrate that national statistics on NBDs in LMICs could be improved by integrating information from readily available databases [15]. In 2023, collaborating with Health Administration Division, MoPH Thailand, we implemented AMASSv2.0 in 127 public hospitals in Thailand using the data from 2022 [19]. We recently reported the epidemiology of AMR bloodstream infections in 111 hospitals [20].

Here we aimed to evaluate the epidemiology of multiple NBDs in 111 hospitals in Thailand using the data from 2022. We also examined the trends of each NBD by comparing the data from 2022 with the data from 2012 to 2015.

## Methods

### Study setting

In 2022, Thailand had a population of 66.1 million, consisted of 77 provinces, and covered 513,120 km². The health systems in each province were integrated into 12 groups of provinces, known as health regions, plus the capital Bangkok as health region 13 (Fig 1), using the concept of decentralization [21]. The Health Administration Division, Ministry of Public Health (MoPH) Thailand, supervised 127 public referral hospitals in health regions 1 to 12. These included 35 advanced-level referral hospital (i.e., level A, with a bed size of about 500-1,200 beds), 55 standard-level referral hospital (i.e., level S, with a bed size of about 300 to 500) and 37 mid-level referral hospital (i.e., level M1, with a bed size of about 180-300) [22]. All level A and S hospitals, and most of level M1 hospitals were equipped with a microbiology laboratory capable of performing bacterial culture using standard methodologies for bacterial identification and susceptibility testing provided by the Department of Medical Sciences, MoPH, Thailand [23]. Public referral hospitals in Bangkok (health region 13) were managed under Department of Medical Services, MoPH (i.e., not under the Health Administration Division, MoPH). Therefore, public referral hospitals in Bangkok were not included, and their summary data were neither generated nor accessed in the study.

From 16 December 2022 to 30 June 2023, on behalf of the Health Administration Division, MoPH, we invited and trained 127 public referral hospitals in health regions 1 to 12 to utilize

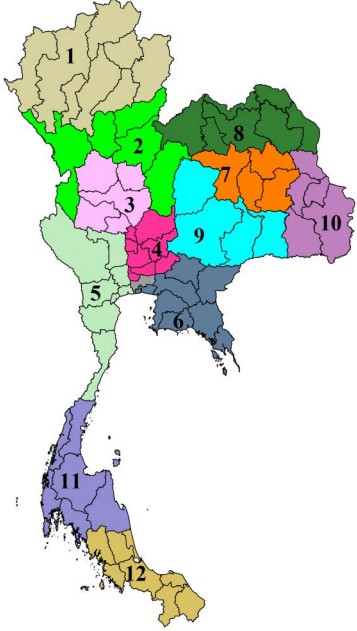

**Fig 1. Health regions in Thailand.** National Health Security Office (NHSO) Region 1 is in Chiang Mai, 2 in Phitsanulok, 3 in Nakhonsawan, 4 in Saraburi, 5 in Ratchaburi, 6 in Rayong, 7 in Khon Kaen, 8 in Udon Thani, 9 in Nakonratchasima, 10 in Ubonratchathani, 11 in Suratthani, 12 in Songkhla and 13 in Bangkok. Map created by the authors using STATA version 14.2 (StataCorp, College Station, TX) and a base layer map from GADM (https://gadm.org/download_country.html) (term of use: https://gadm.org/license.html).

the AMASS with their own microbiology and hospital admission data files via four online meetings, five face-to-face meetings and on-line support [20]. Subsequently, the hospitals that completed utilization of the AMASS submitted summary data generated by the tool to the MoPH [20]. Data were accessed for research purposes from 15 December 2023 to 31 January 2024.

## Study design

We conducted a retrospective study evaluating epidemiology of selected NBDs diagnosed by culture using microbiology laboratory and hospital admission data from 2022. We also compared the yearly incidence rates of each NBD in 2022 with those from 2012-2015, using paired data from 49 public referral hospitals that were previously published [15].

The NBDs under evaluation included infections caused by 11 pathogens; *Brucella* spp., *Burkholderia pseudomallei, Corynebacterium diphtheriae, Neisseria gonorrhoeae, Neisseria meningitidis,* Non-typhoidal *Salmonella* spp., *Salmonella enterica* serovar Paratyphi, *Salmonella enterica* serovar Typhi, *Shigella* spp., *Streptococcus suis*, and *Vibrio* spp. infections. NBD cases were defined as having any clinical specimen (including blood, respiratory tract specimens, urine, stool, cerebrospinal fluid, genital swabs and other specimens) culture positive for a pathogen. The AMASS merged microbiology laboratory and hospital admission data using the hospital number (i.e., the patient identifier) present in both data files [15,20]. Then, the AMASS deduplicated the data and reported total number of inpatients with a clinical specimen culture positive for a pathogen during the evaluation period. Mortality was defined using the discharge summary (in the hospital admission data) which was routinely completed by the attending physician and reported to the MoPH. For each NBD, in case that a patient was admitted with that NBD more than once during the evaluation period, the mortality outcome of the first admission was presented. This approach was taken to avoid overestimating mortality due to other causes (e.g., healthcare-associated infections and road accidents) that might be associated with the subsequent admissions. Some patients with certain NBDs (e.g., *B. pseudomallei* infection) may need several months of treatment to achieve culture negativity.

## Statistical analysis

Data were summarized with medians and interquartile ranges (IQR) for continuous measures, and proportions for discrete measures. IQRs are presented in terms of 25th and 75th percentiles. Continuous variables and proportions were compared between groups using Kruskal Wallis tests and Chi-square tests, respectively.

For NBDs with more than 100 cases in the year 2022, we evaluated factors associated with the incidence rate of NBDs per 100,000 population using multivariable Poisson random-effects regression models [24]. Our summary data from each hospital could be considered nested within its each province. The Poisson random-effects models are appropriate for count outcomes and allow the analysis to partition the total variation in the outcome into between-cluster variation and between-individual variation [24]. The magnitude of the effect of clustering provides a measure of the general contextual effect. The total number of NBD cases in each province was calculated by summing the number of NBD cases from all hospitals located in the same province. We assumed that the distribution of province-specific random effects was normal. Factors evaluated included health region, Gross Provincial Product (GPP), pig density and poultry density. Data of GPP in 2021 [25] were used as a proxy for the size of the economy in each province. Pig density and poultry density (per square meters) were estimated by using the total number of pigs and poultry in each province in Thailand in 2022, divided by the total area of each province [26].

Additionally, we compared the yearly incidence rates of each NBD in 2022 with the those between 2012-2015 in hospitals where paired data were available. The previous study [15] and the period between 2012-2015 were chosen as the comparator because it was the only study that collected both microbiology and hospital admission data and used the same analytical approach for the 11 NBDs. Specifically, AMASS was also previously utilized to merge both data files, deduplicate the data, and report total number of inpatients with a clinical specimen culture positive for a pathogen during the evaluation period [15]. Multivariable Poisson random-effects regression models were utilized to evaluate the change of yearly incidence rate per 100,000 population between the time periods.

We also compared the total number of cases and deaths of each NBD diagnosed by culture in 2022 in our study with those of relevant notifiable diseases reported to the national surveillance systems of Thailand in 2022 [27]. We used STATA (version 14.2; College Station, Texas) for all analyses (S1 Text).

## Ethics

Ethical permission for this study was obtained from the Institute for the Committee of the Faculty of Tropical Medicine, Mahidol University (TMEC 23-085). Individual patient consent was not sought as this was a retrospective study, and the Ethical and Scientific Review Committees approved the process.

## Results

### Baseline characteristics

Of 127 public referral hospitals, 116 (91%) used the AMASS to analyze their microbiology and hospital admission data files, and submitted the summary AMR and NBD data from 2022 to the MoPH. Four hospitals had incomplete microbiology data, and one hospital had incomplete hospital admission data. These hospitals were removed from the analysis. Therefore, a total of 111 hospitals were included in the final analysis.

Of all public referral hospitals in Thailand, 100% of Level A hospitals (35/35), 89% of Level S hospitals (49/55) and 73% of Level M1 hospitals (27/37) were included in this study. Data were available from 74 of 77 provinces (96%) in Thailand, all provinces except Mae Hong Son, Nakorn Nayok and Bangkok.

A total of 46 hospitals in 42 provinces had paired data from 2022 and 2012-2015 (Fig S1). The 46 hospitals were located in all 12 health regions. There was no difference in proportions of health regions among hospitals with and without paired data (p=0.31). Nineteen hospitals were Level A (41%), 20 Level S (44%) and 7 Level M1 (15%). There was borderline evidence showing that the proportion of Level M1 among hospitals with paired data was lower than those without paired-data (15% [7/46] vs. 31% [20/65], p=0.079).

### *Brucella* spp.

In 2022, there were 11 cases with culture-confirmed *Brucella* spp. infection (Fig 2A) and 1 of them died (in-hospital mortality 9%). Among provinces where paired data were available, the yearly incidence rate in 2022 did not differ from that between 2012-2015 (p=0.15).

### B. pseudomallei

In 2022, there were 4,407 cases with culture-confirmed *B. pseudomallei* infection and 1,219 of them died (in-hospital mortality 27.7%). The incidence rate (Fig 2B) was highest in the northeast (health regions 7, 8, 9 and 10), followed by the upper central (health region 3),

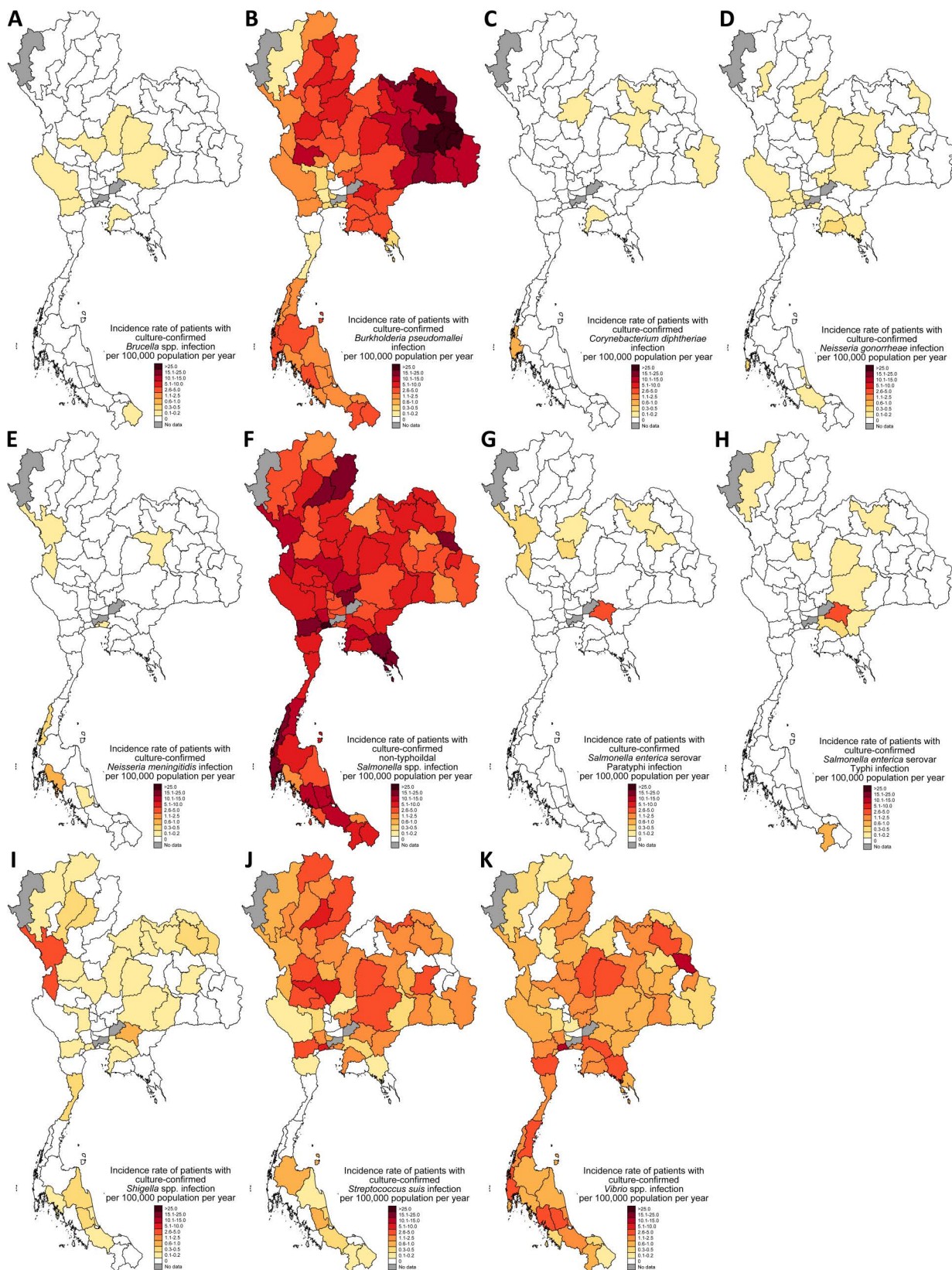

**Fig 2. Incidence rate of cases with notifiable bacterial diseases diagnosed by culture per 100,000 population in 2022 in Thailand. A.** *Brucella* spp. infection, **B.** *Burkholderia pseudomallei* infection, **C.** *Corynebacterium diphtheriae* infection, **D.** *Neisseria gonorrhoeae* infection, **E.** *Neisseria meningitidis* infection, **F.** Non-typhoidal *Salmonella* spp. infection, **G.** *Salmonella enterica* serovar Paratyphi infection, **H.** *Salmonella*

*enterica* serovar Typhi infection, **I.** *Shigella* spp. infection, **J.** *Streptococcus suis* infection, and K. *Vibrio* spp. infection. Map created by the authors using STATA version 14.2 (StataCorp, College Station, TX) and a base layer map from GADM (https://gadm.org/download_country. html) (term of use: https://gadm.org/license.html).

north (health region 1 and 2), east (health region 6) and south (health region 11 and 12). The incidence rate was lowest in the lower central (health region 4) and west (health region 5). In the multivariable models, health region and poultry density (adjust incidence rate ratio [aIRR] 1.31, 95%CI 1.02-1.69, p=0.04) were independently associated with the incidence of *B. pseudomallei* infection per 100,000 population (Table S1). GPP and pig density were not independently associated with the incidence rate (both p>0.20)

Among provinces where paired data were available, the yearly incidence rate of *B. pseudomallei* infection in 2022 was higher than that between 2012-2015 by 58% (aIRR 1.58, 95%CI 1.49-1.68, p<0.001).

### C. diphtheriae

In 2022, there were 10 cases with culture-confirmed *C. diphtheriae* infection (Fig 2C) and 1 of them died (in-hospital mortality 10%). Among provinces where paired data were available, the yearly incidence rate in 2022 did not differ from that between 2012-2015 (p=0.29).

### N. gonorrhoeae

In 2022, there were 25 cases with culture-confirmed *N. gonorrhoeae* infection (Fig 2D) and no deaths following the infections were observed. Among provinces where paired data were available, the yearly incidence rate in 2022 did not differ from that between 2012-2015 (p=0.33).

### N. meningitidis

In 2022, there were 9 cases with culture-confirmed *N. meningitidis* infection (Fig 2E) and 2 died (in-hospital mortality 22%). Among provinces where paired data were available, the yearly incidence rate in 2022 did not differ from that between 2012-2015 (p=0.55).

### Non-typhoidal *Salmonella* spp. (NTS)

In 2022, there were 4,501 cases with culture-confirmed NTS infection (Fig 2F) and 461 died (in-hospital mortality 10.2%). In the multivariable models, health region, GPP, pig density and poultry density were not associated with the incidence rate (all p>0.20, Table S2).

Among provinces where paired data were available, the yearly incidence rate of NTS cases in 2022 was lower than that between 2012-2015 by 37% (aIRR 0.63, 95%CI 0.60-0.67, p<0.001).

### *Salmonella enterica* serovar Paratyphi

In 2022, there were 30 cases with culture-confirmed *Salmonella enterica* serovar Paratyphi infection (Fig 2G) and 4 died (in-hospital mortality 13%). Among provinces where paired data were available, the yearly incidence rate in 2022 did not differ from that between 2012-2015 (p=0.39).

### *Salmonella enterica* serovar Typhi

In 2022, there were 32 cases with culture-confirmed *Salmonella enterica* serovar Typhi infection (Fig 2H) and 6 died (in-hospital mortality 19%).

Among provinces where paired data were available, the yearly incidence rate of *Salmonella enterica* serovar Typhi infection in 2022 was lower than that between 2012-2015 by 83% (aIRR 0.17, 95%CI 0.07-0.41, p<0.001).

### *Shigella* spp.

In 2022, there were 68 cases with culture-confirmed *Shigella* spp. infection (Fig 2I) and 4 died (in-hospital mortality 6%).

Among provinces where paired data were available, the yearly incidence rate of *Shigella* spp. infection in 2022 was lower than that between 2012-2015 by 78% (aIRR 0.22, 95%CI 0.14-0.36, p<0.001).

### *S. suis*

In 2022, there were 867 cases with culture-confirmed *S. suis* infection and 134 of them died (in-hospital mortality 15.5%). The incidence rate (Fig 2J) was highest in the upper central (health regions 3), followed by the north (health regions 1 and 2) and northeast (health regions 7, 8, 9 and 10). The incidence rate was lowest in the south (health regions 11 and 12) and lower central (health region 4). In the multivariable models, health region was associated with the incidence of *S. suis* infection per 100,000 population (Table S3). GPP, pig density and poultry density were not associated with the incidence rate (all p>0.10).

Among provinces where paired data were available, the yearly incidence rate of *S. suis* infection in 2022 was higher than that between 2012-2015 by 172% (aIRR 2.72, 95%CI 2.29-3.24, p<0.001).

### *Vibrio* spp.

In 2022, there were 809 cases with culture-confirmed *Vibrio* spp. infection and 122 of them died (in-hospital mortality 15.1%). The incidence rate (Fig 2K) was highest in the west (health region 5), followed by the east (health region 6). The incidence rate was lowest in the central (health region 3 and 4) and upper north (health region 1). In the multivariable models, health region was associated with the incidence of *Vibrio* spp. infection per 100,000 population (Table S4). GPP, pig density and poultry density were not independently associated with the incidence rate (all p>0.10).

Among provinces where paired data were available, the yearly incidence rate of *Vibrio* spp. infection in 2022 was lower than that between 2012-2015 by 25% (aIRR 0.75, 95%CI 0.66-0.85, p<0.001).

### Comparison with the national surveillance systems

In 2022, the total number of cases with *B. pseudomallei* infection diagnosed by culture (4,407 cases) was higher than that reported to the national surveillance systems (3,357 cases, a 1.3-fold difference, Table 1). The total number of deaths following *B. pseudomallei* infection diagnosed by culture (1,129 deaths) was higher than that reported to the national surveillance systems (157 deaths, a 7.2-fold difference). The total number of cases with *S. suis* infection diagnosed by culture (867 cases) was also higher than those reported to the national surveillance systems (383 cases, a 2.3-fold difference). The total number of deaths following *S. suis* infection diagnosed by culture (134 deaths) was also higher than that reported to the national surveillance systems (10 deaths, a 13.4-fold difference). The total number of deaths following fecal-oral transmitted NBDs diagnosed by culture (including 461 for non-typhoidal salmonella, 122 for vibriosis, 4 for paratyphoid, 6 for typhoid and 4 for shigellosis) were also higher than those reported to the national surveillance systems (0 deaths for all relevant fecal-oral transmitted NBDs).

**Table 1. Total number of cases and deaths following selected notifiable bacterial diseases (NBDs) diagnosed by culture in 111 hospitals compared with total number of cases and deaths with relevant notifiable diseases reported to the national surveillance systems (NSS) in Thailand in 2022.**

| Infections* | This study | | NSS | |
|---|---|---|---|---|
| | Cases | Deaths | Cases | Deaths |
| *Brucella* spp. infection | 11 | 1 | 15 | 0 |
| *Burkholderia pseudomallei* infection | 4,407 | 1,219 | 3,573 | 157 |
| *Corynebacterium diphtheriae* infection | 10 | 1 | 0 | 0 |
| *Neisseria gonorrhoeae* infection | 25 | 0 | 6,915 | 0 |
| *Neisseria meningitidis* infection | 9 | 2 | 19 | 3 |
| Non-typhoidal *Salmonella* spp. infection | 4,501 | 461 | 72,439 | 0 |
| *Salmonella enterica* serovar Paratyphi infection | 30 | 4 | 91 | 0 |
| *Salmonella enterica* serovar Typhi infection | 32 | 6 | 754 | 0 |
| *Shigella* spp. infection | 68 | 4 | 377 | 0 |
| *Streptococcus suis* infection | 867 | 134 | 383 | 10 |
| *Vibrio* spp. infection | 809 | 122 | 4 | 0 |

*In this study, cases were defined as an inpatient with a clinical specimen culture positive for a pathogen during the evaluation period, while the NSS included reports of suspected, probable and confirmed cases using a wide range of case definitions of each notifiable disease (S2 Text).

## Discussion

Our findings provide evidence that the incidences of melioidosis and *S. suis* infections in Thailand in 2022 are increasing compared to the data from 2012-2015 and associated with a high number of deaths. We also show that fecal-oral transmitted NBDs including non-typhoidal salmonellosis, typhoid, shigellosis and vibriosis are still present and associated with deaths, but their incidence rates are decreasing compared to the data from 2012-2015. This study highlights the potential advantage of utilization of routine microbiology and hospital admission data from hospitals. The local and timely data of NBDs can supplement and monitor the performance of the national surveillance systems. The accurate data can consequently identify diseases and areas with high burden, improve public health interventions, and prioritize resource allocation.

Underreporting of cases and deaths following *B. pseudomallei*, *S. suis* and other infections to the national surveillance systems is consistent with the previous finding [11,16]. It is likely that many healthcare workers responsible for reporting to the national surveillance systems are still unaware that all cases and deaths following culture-positive NBDs must be reported [16] and that ICD-10 of those culture-confirmed cases in the hospitals' health information systems were largely inaccurate [13,14]. Additionally, their perceptions of their workload and the belief that reporting cases is not used for outbreak response may contribute to underreporting [28]. Further studies on underreporting and on automating reporting systems utilizing laboratory data from healthcare facilities are required.

The finding that there were more than 1,200 deaths following melioidosis in 2022 is alarming. This finding is consistent with a previous modelling study predicting that the total number of deaths following melioidosis could range from 1,259 to 6,678 in Thailand if all patients were tested with bacterial culture and data were reported nationwide [29]. Melioidosis is usually acquired by skin inoculation, ingestion or inhalation of *B. pseudomallei* in soil and water. Preventive measures include wearing appropriate protective clothing and footwear when contacting with soil or water, consuming bottled or boiled water, and ensuring that the domestic water supply is not contaminated with *B. pseudomallei* [30–32]. The increased incidence rate of melioidosis could be associated with the increasing incidence of diabetes (the major risk factor of melioidosis) and improvements in diagnostic stewardship (i.e., utilization

of culture) and bacterial identification in public hospitals in Thailand over time [16,33,34]. The observed increase was not due to the differences in the geographical regions evaluated in 2022 compared to 2012-2015, as we evaluated only provinces with paired data. The difference in the incidence by regions in Thailand is consistent with previous studies [29]. The association between poultry density and melioidosis has never been observed, and could be due to a collinearity with residual confounding factors (e.g., agricultural occupation). Further studies and actions to reduce the burden of melioidosis in Thailand are urgently needed [35].

Similarly, the increase in *S. suis* infection is alarming. Risk factors for *S. suis* infection include consuming undercooked pork contaminated with *S. suis* and exposure to pigs or pork. The increased incidence could be associated with an increase in consumption of undercooked pork products [36–38] and an increase in infected meat in the market [39]. The latter is a concern following the news of the large illegal pork imported to Thailand since 2021 after the shortage of domestic pork due to the outbreak of African swine fever in Thailand [40]. The increased incidence rate of *S. suis* infection could partially be associated with improvements in diagnostic stewardship and bacterial identification in public hospitals in Thailand over time. Nonetheless, these factors alone cannot explain the nearly three-fold increase in *S. suis* infection in the country. The DDC will utilize the data to additionally implement and enforce behavioral interventions, education and food biosafety in the country [38]. These interventions include educating at-risk populations, such as pig farmers, and increasing public disease awareness and prevention efforts through public health officers and village health volunteers. Additionally, food biosafety measures will be enforced in collaboration with relevant stakeholders, emphasizing proper handling and processing of pork at abattoir, market and consumer levels to prevent contamination and transmission.

There are currently no licensed vaccines for melioidosis, *S. suis* infection and shigellosis. Vaccination for salmonella and vibriosis are not widely available and are seldom used in Thailand.

The striking decrease in incidence rates of *Salmonella enterica* serovar Typhi and *Shigella* spp. infections (83% and 78% decrease, respectively) could be due to improvement in clean water supply, sanitation, regulatory environment, food safety, and related health intervention programs over time [41,42]. Improvement in health-seeking behaviour and access to healthcare under the universal coverage have likely contributed to the decrease [43,44]. Although the incidence rates of NTS and *Vibrio* spp. infection minimally decrease (37% and 25% decrease), these pathogens still cause substantial public health problems across the regions. The higher incidence rate of NTS infection compared to typhoid is consistent with those observed in other LMICs [45]. Persistence of virulent or drug-resistant strains among susceptible population including children and people with co-morbidities may be associated with the slow reduction of the NTS [45] and *Vibrio* spp. infections [46]. The MoPH and related stakeholders should maintain and strengthen the public health interventions to decrease incidences of these infections further. Further studies and measures to reduce the burden of NTS and *Vibrio* spp. infections are also required.

Our study and approach have several strengths. First, our study included most of the public referral hospitals in the country. Second, hospital admission data used in the analysis (including hospital number, admission date, discharge date and in-hospital mortality) were reliable, as these data were recorded electronically and required for retrospective hospital payment nationwide [47,48]. Third, we utilized microbiology laboratory data, which is highly specific to the diagnosis of NBDs. Fourth, the method for collecting data between 2012 and 2015 was similar to those used in 2022 [15], utilizing microbiology and hospital admission data and AMASS. Fifth, although our approach is semi-automatic, this approach is easy to scale up in LMICs because the AMASS programme is open-access, highly-compatible, and user-friendly without the need for data experts with adequate skills in statistical software [17,19].

Our study and approach have several limitations. First, our approach included only inpatients in the public referral hospitals. Therefore, our estimates did not include patients who did not require hospitalization, and those who were hospitalized in private, military or university hospitals. According to data from the Healthcare Accreditation Institute, Thailand [49], the public referral hospitals included in this study represented 56.6% (120/212) of hospitals with at least 180 beds, 66.4% (81/122) of hospitals with at least 300 beds and 70% (42/60) of hospitals with at least 500 beds in Thailand. Second, our approach is not applicable in settings where microbiology and hospital admission data are not computerized. Third, our approach focused on bacterial culture results. Therefore, the findings could be influenced by diagnostic stewardship and, capability and expertise of the microbiology laboratories. Methods used for bacterial culture and identification may differ by region and over time. For example, some hospitals might have implemented new bacterial identification automated systems in 2022, reducing the likelihood of misidentification of *S. suis* (as *Streptococcus* spp.) and *B. pseudomallei* (as contaminants or *Pseudomonas* spp.). Fourth, our approach utilizes summary data and could not evaluate individual-level clinical data and microbiology laboratory data in details. Fifth, the AMASS did not generate reports for patients who were culture positive for more than one pathogen. Such patients would be counted separately for each pathogen. Sixth, the AMASS did not generate reports for number of admissions among patients who were culture positive for pathogens under evaluation. Seventh, the all-cause mortality could be higher than in-hospital mortality because a preference to die at home is high in some regions in Thailand [11].

In conclusion, the burden of melioidosis and *S. suis* infections is increasing in Thailand. Their incidence rates are higher in some regions than in others. Although the incidence rates of NTS infection and vibriosis are decreasing, they are still associated with a high number of cases and deaths. Specific public health interventions to reduce the burden of melioidosis, *S. suis* infection, NTS infection and vibriosis are urgently required.

## Supporting information

**S1 Table. Factors associated with the incidence of cases with culture-confirmed *Burkholderia pseudomallei* infection per 100,000 population in 74 provinces in Thailand, 2022**
(Word)

**S2 Table. Factors associated with the incidence of cases with culture-confirmed non-typhoidal *Salmonella* spp. (NTS) infection per 100,000 population in 74 provinces in Thailand, 2022**
(Word)

**S3 Table. Factors associated with the incidence of cases with culture-confirmed *S. suis* infection per 100,000 population in 74 provinces in Thailand, 2022**
(Word)

**S4 Table. Factors associated with the incidence of cases with culture-confirmed *Vibrio* spp. infection per 100,000 population in 74 provinces in Thailand, 2022**
(Word)

**S5 Table. Total number of cases following selected notifiable bacterial diseases (NBDs) diagnosed by culture in 2022 compared with the yearly incidence between 2012-2015 in 49 hospitals where paired data were available**
(Word)

**S1 Text. Stata code for fitting the multivariable Poisson regression models**
(Word)

**S2 Text. Example of reporting criteria of relevant notifiable diseases in Thailand** (Word)

**S1 Fig. Map of 42 provinces that had paired data from 2022 and 2012-2015** . Map created by the authors using STATA version 14.2 (StataCorp, College Station, TX) and a base layer map from GADM (https://gadm.org/download_country.html) (term of use: https://gadm.org/license.html)
(Image)

## Acknowledgement

We gratefully acknowledge the laboratory team and IT team of all hospitals for their participation and support.

## Author contributions

**Conceptualization:** Charuttaporn Jitpeera, Somkid Kripattanapong, Pawinee Doungngern, Papassorn Pinyopornpanish, Panida Chamawan, Voranadda Srisuphan, Krittiya Tuamsuwan, Phairam Boonyarit, John Stelling, Paul Turner, Direk Limmathurotsakul.

**Data curation:** Charuttaporn Jitpeera, Somkid Kripattanapong, Preeyarach Klaytong, Krittiya Tuamsuwan, Direk Limmathurotsakul.

**Formal analysis:** Charuttaporn Jitpeera, Direk Limmathurotsakul.

**Funding acquisition:** Direk Limmathurotsakul.

**Methodology:** Preeyarach Klaytong, John Stelling, Paul Turner, Direk Limmathurotsakul.

**Project administration:** Direk Limmathurotsakul.

**Software:** Chalida Rangsiwutisak, Prapass Wannapinij.

**Supervision:** Pawinee Doungngern, Papassorn Pinyopornpanish, Panida Chamawan, Voranadda Srisuphan, Phairam Boonyarit, Orapan Sripichai, Soawapak Hinjoy, John Stelling, Paul Turner, Wichan Bhunyakitikorn, Sopon Iamsirithaworn, Direk Limmathurotsakul.

**Validation:** Direk Limmathurotsakul.

**Visualization:** Direk Limmathurotsakul.

**Writing – original draft:** Charuttaporn Jitpeera, Direk Limmathurotsakul.

**Writing – review & editing:** Charuttaporn Jitpeera, Somkid Kripattanapong, Preeyarach Klaytong, Chalida Rangsiwutisak, Prapass Wannapinij, Pawinee Doungngern, Papassorn Pinyopornpanish, Panida Chamawan, Voranadda Srisuphan, Krittiya Tuamsuwan, Phairam Boonyarit, Orapan Sripichai, Soawapak Hinjoy, John Stelling, Paul Turner, Wichan Bhunyakitikorn, Sopon Iamsirithaworn, Direk Limmathurotsakul.

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
