## [Decision Letter · Decision Letter 0]

10 Oct 2024

PGPH-D-24-02039

Epidemiology of Burkholderia pseudomallei, Streptococcus suis, Salmonella spp., Shigella spp. and Vibrio spp. infections in 111 hospitals in Thailand, 2022

Dear Dr. Limmathurotsakul,

Thank you for submitting your manuscript to PLOS Global Public Health. After careful consideration, we feel that it has merit but does not fully meet PLOS Global Public Health’s publication criteria as it currently stands. Therefore, we invite you to submit a revised version of the manuscript that addresses the points raised during the review process.

The reviewers have shared major and minor comments to be addressed in the updated manuscript. In particular, additional context should be provided about the study that was conducted from 2012-2015 and several of the pathogen-specific findings from that study, including how incidence rates were calculated and what any reported changes might be attributable to. How are findings from these two studies and study periods comparable? Please also explain why there are no data included from Bangkok in this analysis. 

We look forward to receiving your revised manuscript.

Kind regards,

Megan Elizabeth Carey

Academic Editor

Journal Requirements:

**Please only choose the relevant sentences from below**

1) Please clarify all sources of funding (financial or material support) for your study. List the grants (with grant number) or organizations (with url) that supported your study, including funding received from your institution. 

2) State the initials, alongside each funding source, of each author to receive each grant.

3) State what role the funders took in the study. If the funders had no role in your study, please state: “The funders had no role in study design, data collection and analysis, decision to publish, or preparation of the manuscript.”

4) If any authors received a salary from any of your funders, please state which authors and which funders.

2. We have noticed that you have uploaded Supporting Information files, but you have not included a list of legends. Please add a full list of legends for your Supporting Information files after the references list. 

3. Figures 1 and 2: please (a) provide a direct link to the base layer of the map (i.e., the country or region border shape) and ensure this is also included in the figure legend; and (b) provide a link to the terms of use / license information for the base layer image or shapefile. We cannot publish proprietary or copyrighted maps (e.g. Google Maps, Mapquest) and the terms of use for your map base layer must be compatible with our CC-BY 4.0 license. 

Additional Editor Comments (if provided):

Reviewers' comments:

Reviewer's Responses to Questions

**Comments to the Author**

1. Does this manuscript meet PLOS Global Public Health’s publication criteria? Is the manuscript technically sound, and do the data support the conclusions? The manuscript must describe methodologically and ethically rigorous research with conclusions that are appropriately drawn based on the data presented.

Reviewer #1: Yes

Reviewer #2: Partly

2. Has the statistical analysis been performed appropriately and rigorously?

Reviewer #1: I don't know

Reviewer #2: Yes

3. Have the authors made all data underlying the findings in their manuscript fully available (please refer to the Data Availability Statement at the start of the manuscript PDF file)?

Reviewer #1: Yes

Reviewer #2: Yes

4. Is the manuscript presented in an intelligible fashion and written in standard English?

Reviewer #1: Yes

Reviewer #2: Yes

5. Review Comments to the Author

**Reviewer #1**

Overall comments

Jitpeera et al. analyse the incidence of 11 different bacterial pathogens from data they have collected from the majority of Thailand’s hospitals in 2022, and then compare incidence from some of these regions with paired data from 2012-2015. In-hospital mortality and associations with local agricultural animal density, economic data and the health region are also investigated. The scale of the data and observed trends are valuable to the field and overall the data is well presented and described, and conclusions are appropriately supported by the data. However, several points need to be clarified and some recommendations for additional data presentation are included below. While the discussion raises good points, this section would benefit from expanding slightly to put more details into context, particularly for readers not familiar with these bacterial pathogens. Unable to comment on the validity of the statistical analysis.

Major comments

1. Line 47. At some point in the manuscript (methods?) the authors should explain why there is no data from Bangkok.

2. Line 58. “The incident rate of NTS infection was not associated with geographical region.” This sentence is not accurate as written, and should be rephrased to make it clear that NTS incidence does vary by region, perhaps by as much as 10-20 times (from the data shown in Fig 2F). I believe the authors meant to say that NTS incidence was generally high across all regions of Thailand, and there is no clear epicentre of infection.

3. Line 59 – 61. The authors highlight that the incidence fecal-oral transmitted NBDs was lower and that of B. pseudomallei and S. suis was higher. The route of transmission of the later should be mentioned (perhaps in the discussion).

4. Line 78-81. It should be made clear whether any of the 13 dangerous communicable diseases are bacterial and if so what they are.

5. Line 84: It should be noted whether the ICD-10 is reliable for any of the 11 bacterial diseases described in the manuscript. The following sentence stating that microbiology lab data is not used would suggest that they are not, but this should be made clear.

6. Line 119-121 & Line 139 - 142: Some more specifics on how each bacterial disease was cultured and identified is warranted. Ref 23 handbook link is broken, but is dated as 2012; has there been any update to procedures/methods since? It should be made clear whether the methods for collecting data differ between the earlier timepoint and the later. This is particularly important for Burholderia and S. suis which have seen large increases in incidence over the period.

7. While culture is the gold standard for the diagnosis of many of these pathogens, the authors should note in the discussion if there are any limitations with using solely culture as a readout for specific pathogens that may bias the data. This is very briefly mentioned in the discussion at line 344-346, but more detail should be included in terms of how cultured methods may vary between regions if know.

8. Were any patients culture positive for more than one pathogen? Regardless of answer this should be stated if known.

9. Line 146-147: How many patients had multiple admissions? Explain why mortality data was based only on the first admission.

10. Line 165-166. It should be noted what biases the paired data introduce in the analysis. Eg. what regions/ hospital types were less/more represented in the 2012-2015 dataset relative to that of the 2022 dataset.

11. Line 196: Which regions are included in the 2012-2015 dataset? Can this be displayed on a map in a figure?

12. Line 288: This paragraph compares the collected data with that of another dataset from the national surveillance system, and concludes that there are significant differences between the two datasets in terms of incidence and mortality. The argument is made in the discussion in lines 305 -306 that “local data can supplement and monitor the performance of the national surveillance systems”. However, it is not explained why such a disparity exists between the two datasets, and it is not obvious to the reader why; this needs to be clearly explained in the discussion.

13. Line 320: specify “increase from 2012-2015 to 2022”. The authors should note, as they have done for burkholderia, whether the increase observed for S. suis could have been caused by changes in diagnostic procedures/methods etc.

14. Line 327: specify the “fecal-oral transmitted diseases”. Transmission of burkholderia and S. suis should be discussed as a point of difference.

15. Line 328-330: This sentence is very general. Be specific about what interventions could decrease incidence and why.

16. Line 351: In the concluding paragraph only melioidosis and S. suis are mentioned as requiring specific public health inventions. While it is true that these have increased dramatically and have the highest associated mortality, the data shows that NTS and cholera, despite decreasing over the same period, continue to have high incidence in Thailand and high mortality. Therefore NTS and cholera should be highlighted as continuing to have high burden in the country and requiring specific public health interventions.

17. The comparison is between a multiyear timeframe (2012-2015) and a single year. Was there anything different or special about the 2022 year (eg. climate) that may have lead to a changes in disease transmission? Have any vaccination programs/antimicrobial treatment/general healthcare programs for any of these pathogens been introduced since 2015 which may have affected disease incidence?

18. Is there patient meta data available that could help explain changes in incidences and affected populations eg. age, gender, comorbidities, coinfections with more than one NBD?

Figures/Tables:

1. Figure 2:

a. The colours at the high end of incidence are similar and hard to differentiate. Recommend adapting the colours (perhaps to a 2 or 3 colour system?) to make the difference incidence levels easier to distinguish.

b. The size of the numbers in the colour legend should be increased to a more legible level.

c. Fig legend should note the pathogen for each of the numbered graphs.

2. The authors compare paired data from 2022 with that of 2012-2015. It should be made clear what regions/provinces are included in the this temporal comparison. Perhaps a map highlighting the regions/provinces incorporated? A geographical figure similar to Fig 2 that shows the % or fold change in incidence of each NBD during this period may work. Or could be a column in a data table.

3. Recommend that the authors include a table summarising all data for easy comparison between pathogens eg. total patients, total deaths, % deaths, % change from 2012-2015 data.

Minor comments

1. Line 135: Should be 11 pathogens, not 12.

2. Ref 23 url does not open.

3. Line 201: “the time periods” should be specified.

4. Line 225: “none died” should be rephrased to “no associated mortality” or similar.

5. Line 289-290. Make clear that the second figures in the brackets are from national surveillance and quantify the difference eg. in fold or %

6. Line 300-301: Make clear that increase is relative to 2012-2015 data.

7. Line 310. Sentence not grammatically correct, rephrase.

8. Line 316-317: “spurious” is not the right word here.

9. Line 332-33: Quantify the % of referral hospitals in the country included in the data.

**Reviewer #2**

The study presents a thorough analysis of notifiable bacterial diseases (NBDs) in Thailand, providing valuable insights into emerging epidemiological trends. By leveraging comprehensive microbiology and hospital admission data from a large network of public referral hospitals, the authors address a significant knowledge gap regarding the incidence and distribution of NBDs in low- and middle-income countries. The use of multivariable Poisson random-effects regression models and the comparison with historical data (2012-2015) strengthen the robustness of the findings. However, there are some major and minor comments and areas that warrant further consideration for clarification and improvement.

**Major comments**

Please summarize the study that was performed in 2012-2015 as this is used as comparator. Why was this period chosen as a comperator? Please explain why chosen for these eleven bacterial diseases. And where these already notifiable pathogens in 2012 for example? 

At the moment, I do not understand what you are comparing your results to and how to interpretate the major increases and decreases in incidence levels. For example, if the regions where melioidosis is known to be highly endemic were not included in the 2012-2015 study, but are now included in the 2022 study due to its larger scope, this obviously explains the increase in incidence. Without this information (or adjustment for this fact) in the analysis, it’s not possible to interpretate the incidence increases or decreases as mentioned below under a to f. It may be true that there’s an increase in awareness and subsequent identification, but then my main question remains: is the incidence increasing or is the detection increasing? The authors should be more critical about this distinction. 

Line 218. The authors state that the incidence of *B. pseudomallei* increased by 58% in 2022 compared to 2012-2015. Can the authors explain this major increase? The reasons mentioned in the discussion might be true (line 318-320), but please incorporate the suggestions above.  Line 242. The authors state that the incidence of NTS decreased by 37% in 2022 compared to 2012-2015. Can the authors explain this major decrease? Please elaborate on this decrease in the discussion. Line 254. The authors state that the incidence of *Salmonella enterica serovar Typhi* decreased by 83% in 2022 compared to 2012-2015. Can the authors explain this major decrease? Please elaborate on this decrease in the discussion.Line 262. The authors state that the incidence of Shigella decreased by 78% in 2022 compared to 2012-2015. Can the authors explain this major decrease? Please elaborate on this decrease in the discussion.Line 275. The authors state that the incidence of *S. suis* increased by 172% in 2022 compared to 2012-2015. Can the authors explain this major increase? The reasons mentioned in the discussion might be true (line 324-328), but please incorporate the suggestions above.  Line 288. The authors state that the incidence of Vibrio decreased by 25% in 2022 compared to 2012-2015. Can the authors explain this major decrease? Please elaborate on this decrease in the discussion. 

Is the use of **multivariable Poisson random-effects regression models** appropriate for this type of data and analysis? Please explain why this model was chosen.The study relies on hospital discharge summaries to define mortality. How confident are you in the reliability and consistency of these summaries across different hospitals and regions? Given the possibility of discrepancies in reporting. Why was Bangkok excluded from the analysis?

**Minor comments**

Line 188. The authors state that 91% used the AMASS to acquire data, but how was the other 9% of the data obtained? Please elaborate. Table 1. Suggestion to place the total number of cases and deaths in columns next to each other. This is easier for the reader to comprehend as it immediately reads as a mortality rate. Also, suggest to use the same nomenclature for the disease/causative agent. So, column 1 and column 3 should be combined to a single column. If there are discrepancies between the definition used in the AMASS and NSS, state so in the methods or supplemental files. Line 331-332. The authors mention that measures were taken over time that resulted in a decrease in fecal-oral transmitted diseases. The information is very brief.

The decrease in fecal-oral transmitted diseases such as typhoid and shigellosis is attributed to improvements in sanitation and clean water supply. Could other factors, such as changes in healthcare-seeking behavior, vaccination campaigns, or environmental factors, play a role in this decline? How did you isolate the specific impact of water and sanitation improvements from these other potential influences?  Please give more information on the context of the interventions. Readers should not have to go back and forth to other publications to identify this information, especially since it is considered critical information by the authors as it explains decreases in incidence levels. 

The study highlights regional differences in the incidence of *B. pseudomallei* and *S. suis* infections. Could these regional differences be influenced by varying healthcare access, diagnostic capacity, or reporting practices across provinces? How did the study address such potential confounding factors to ensure that the differences are truly epidemiological rather than artefacts of the healthcare system?The study calls for public health interventions to address the increasing burden of *melioidosis* and *S. suis* infections. Given the complexity of these diseases, what specific interventions would be both effective and feasible in Thailand's current public health landscape? How would you prioritize resource allocation, and what are the potential barriers to implementing these interventions across diverse regions?While the AMASS program is scalable and user-friendly, how generalizable is the study’s approach to other low- and middle-income countries (LMICs) with different healthcare infrastructures, disease profiles, and data management capabilities? Could the reliance on computerized microbiology and hospital data present a challenge for broader adoption in less digitized health systems?

6. PLOS authors have the option to publish the peer review history of their article (what does this mean?). If published, this will include your full peer review and any attached files.

**Do you want your identity to be public for this peer review?** For information about this choice, including consent withdrawal, please see our Privacy Policy.

Reviewer #1: **Yes: **Timothy A Scott

Reviewer #2: No

---

## [Decision Letter · Decision Letter 1]

1 Jan 2025

PGPH-D-24-02039R1

Epidemiology of Burkholderia pseudomallei, Streptococcus suis, Salmonella spp., Shigella spp. and Vibrio spp. infections in 111 hospitals in Thailand, 2022

Dear Dr. Limmathurotsakul,

Thank you for submitting your manuscript to PLOS Global Public Health. After careful consideration, we feel that it has merit but does not fully meet PLOS Global Public Health’s publication criteria as it currently stands. Therefore, we invite you to submit a revised version of the manuscript that addresses the points raised during the review process.

We look forward to receiving your revised manuscript.

Kind regards,

Megan Elizabeth Carey

Academic Editor

Journal Requirements:

Additional Editor Comments (if provided):

Many thanks for addressing the major review comments from the editors. A few very minor edits still need to be made:

-line 80 - suggest replacing "information" with data

-line 86 - suggest replacing "tuberculosis" with "Mycobacterium tuberculosis" for consistency

-line 90-91 - point about notifiable bacterial diseases warrants additional explanation

-lines 130-2 - thanks for addressing the question about why Bangkok wasn't included, but some additional explanation for why would be welcome (e.g. data could not be accessed, and/or data are captured differently and therefore aren't comparable, etc.)

-line 189 - what is "the same analytical approach"? some additional specificity would be helpful

-line 353 - "underreporting *of* cases and deaths..."

-line 402 - decrease in... numbers? incidence?...of Salmonella enterica serovar Typhi and Shigella spp. infections

-line 403 - decreases*

-line 407 - rates*

-lines 408-410 - what is meant by "burden" - incidence? kind of an imprecise term, could say that these pathogens cause substantial public health problems, or similar

-line 441 - methods *used for*...

-line 442 - some hospitals might have *implemented* ...?

-line 452 - "Although the incidence *rates* of..."

-should be "vibrioses" rather than "fibrosis"

Reviewers' comments:

Reviewer's Responses to Questions

**Comments to the Author**

1. If the authors have adequately addressed your comments raised in a previous round of review and you feel that this manuscript is now acceptable for publication, you may indicate that here to bypass the “Comments to the Author” section, enter your conflict of interest statement in the “Confidential to Editor” section, and submit your "Accept" recommendation.

Reviewer #1: All comments have been addressed

2. Does this manuscript meet PLOS Global Public Health’s publication criteria? Is the manuscript technically sound, and do the data support the conclusions? The manuscript must describe methodologically and ethically rigorous research with conclusions that are appropriately drawn based on the data presented.

Reviewer #1: Yes

3. Has the statistical analysis been performed appropriately and rigorously?

Reviewer #1: I don't know

4. Have the authors made all data underlying the findings in their manuscript fully available (please refer to the Data Availability Statement at the start of the manuscript PDF file)?

Reviewer #1: Yes

5. Is the manuscript presented in an intelligible fashion and written in standard English?

Reviewer #1: Yes

6. Review Comments to the Author

Reviewer #1: Minor: The new ref 23 link is broken

7. PLOS authors have the option to publish the peer review history of their article (what does this mean?). If published, this will include your full peer review and any attached files.

**Do you want your identity to be public for this peer review?** For information about this choice, including consent withdrawal, please see our Privacy Policy.

Reviewer #1: **Yes: **Timothy A Scott

---

## [Editor Report · Decision Letter 2]

16 Jan 2025

Epidemiology of Burkholderia pseudomallei, Streptococcus suis, Salmonella spp., Shigella spp. and Vibrio spp. infections in 111 hospitals in Thailand, 2022

PGPH-D-24-02039R2

Dear Limmathurotsakul,

We are pleased to inform you that your manuscript 'Epidemiology of Burkholderia pseudomallei, Streptococcus suis, Salmonella spp., Shigella spp. and Vibrio spp. infections in 111 hospitals in Thailand, 2022' has been provisionally accepted for publication in PLOS Global Public Health.

Best regards,

Megan Elizabeth Carey

Academic Editor